# High-throughput saturation mutagenesis generates a high-affinity antibody against SARS-CoV-2 variants using protein surface display assay on a human cell

Ye Yang[1]☯, Shuo Liu[2,3]☯, Yufeng Luo[1], Bolun Wang[1], Junyi Wang[1], Juan Li[1], Jiaxin Li[1], Buqing Ye[1], Youchun Wang[3]*, Jianzhong Jeff Xi[1]*

**1** State Key Laboratory of Natural and Biomimetic Drugs, Department of Biomedical Engineering, College of Future Technology, Peking University, Beijing, China, **2** Graduate School of Chinese Academy of Medical Sciences, Peking Union Medical College, Beijing, China, **3** Division of HIV/AIDS and Sex-transmitted Virus Vaccines, Institute for Biological Product Control, National Institutes for Food and Drug Control (NIFDC), Beijing, China

☯ These authors contributed equally to this work.
* wangyc@nifdc.org.cn (YW); jzxi@pku.edu.cn (JJX)

**Data Availability Statement:** Data generated or analyzed during this study are available in the main text or the supplementary materials. DNA

## Abstract

As new mutations continue to emerge, the ability of severe acute respiratory syndrome coronavirus 2 (SARS-CoV-2) virus to evade the human immune system and neutralizing antibodies remains a huge challenge for vaccine development and antibody research. The majority of neutralizing antibodies have reduced or lost activity against SARS-CoV-2 variants. In this study, we reported a novel protein surface display system on a mammalian cell for obtaining a higher-affinity antibody in high-throughput manner. Using a saturation mutagenesis strategy through integrating microarray-based oligonucleotide synthesis and single-cell screening assay, we generated a group of new antibodies against diverse prevalent SARS-CoV-2 variants through high-throughput screening the human antibody REGN10987 within 2 weeks. The affinity of those optimized antibodies to seven prevalent mutants was greatly improved, and the EC50 values were no higher than 5 ng/mL. These results demonstrate the robustness of our screening system in the rapid generation of an antibody with higher affinity against a new SARS-CoV-2 variant, and provides a potential application to other protein molecular interactions.

## Author summary

The COVID-19 pandemic caused by SARS-CoV-2 has become a global health problem of wide concern due to the continuous emergence of mutants, so that human social and economic activities have suffered a serious blow. The development of potent neutralizing antibodies is critically important in the coming time. At present, the main way to obtain effective neutralizing antibodies against SARS-CoV-2 and its variants is to isolate the serum of convalescent patients or immunized humanized mice. However, this method

sequencing data is deposited in publicly available databases (https://ngdc.cncb.ac.cn/gsa-human/browse/HRA003432).

**Funding:** J.X. was supported by the National Key Basic Research Project of China (2018YFA0108101), the National Natural Science Foundation of China (81827809,81421004, T2288102 and 82150005) and Project 2020BD018 supported by PKUBaidu Fund. Y.W was funded by Beijing Municipal Science and Technology Project (Z211100002521018). The funders had no role in study design, data collection and analysis, decision to publish, or preparation of the manuscript.

**Competing interests:** No potential conflict of interest was reported by the authors.

has high requirements on samples, and the whole process is time-consuming and labor-intensive. We here report a single-cell screening assay that allows the rapid and high-throughput optimizing of a human antibody displayed on a mammalian cell surface. The process can improve the affinity of antibodies with EC50 below 5 ng/mL within 2 weeks. Our approach with short turnaround and significant enhancement in antibody affinity presents an effective strategy for fighting against the ever-evolving COVID variants.

## Introduction

Over the past two years, coronavirus disease 2019 (COVID-19) caused by a novel severe acute respiratory syndrome coronavirus (SARS-CoV-2) has spread worldwide, becoming a serious worldwide public health emergency [1, 2]. So far, there have been hundreds of millions confirmed infections and millions of confirmed deaths (World Health Organization) [3]. SARS-CoV-2 with trimeric spike (S) glycoprotein is found to infect human cells by binding to a receptor named angiotensin converting enzyme II (ACE2), and the interaction between S protein and ACE2 was emphasized to be a critical step for viral entry and infection [4, 5]. Thus, monoclonal antibodies (mAbs), most of which target the receptor binding domain (RBD) of the S glycoprotein of SARS-CoV-2 and block the binding between the S protein and the host receptor ACE2, have already shown therapeutic and clinical value against SARS-CoV-2 infection [6, 7]. To date, several hundred neutralizing mAbs from individuals infected with SARS-CoV-2 have been isolated and identified, and some mAbs have been approved for emergency use authorization (EUA) or under active clinical development [8–13]. As the epidemic progress, mutated strains with immune escape ability are emerging and have become less sensitive or completely escaped from antibodies. Five of those mutants were classified as variants of concern (VOC) due to their greater influence, namely Alpha, Beta, Gamma, Delta and Omicron. Some neutralizing antibodies, such as sotrovimab, were slightly less active against Alpha variant due to N501Y. The neutralizing activity of bamlanivimab and casirivimab against Beta variants was completely or significantly lost due to E484K and K417N mutations. The Gamma mutant escaped etesivimab by carrying K417T. Bamlanivimab lost its antiviral activity against the Delta variant that carries L452R. Strikingly, the majority of EUA-approved or advanced clinical development therapeutic neutralizing antibodies failed to neutralize the Omicron mutant due to the presence of 34 mutations, 15 of which are located in the RBD region [14–23].

The generation of antibodies to SARS-CoV-2 and its variant strains with high affinity for clinical use is an important and urgent task. However, the isolation and identification of new antibodies with high affinity from infected individuals is time-consuming and laborious. Alternatively, protein engineering is another good choice to generate an antibody with high activity against a new SARS-CoV-2 variant. In previous studies, phage display and yeast cell surface display systems have been used for the in vitro screening of an antibody with higher affinity [24–32]. However, all of those techniques have been partially successful, since phages and yeast cells are highly different from mammalian cells in terms of intracellular protein folding, post-translational modifications, and codon usage. Therefore, serious problems related to antibody activity, such as different glycosylation modifications that may dramatically change the features and functions of antibodies, can occur when proteins screened in bacterial or yeast systems are transferred to mammalian expression systems [33–36]. Another shortcoming of the in vitro antibody optimization in previous studies resulted from antibody mutant libraries *per se*. The libraries, which were generally generated by error-prone and randomized synthetic

oligonucleotide PCR-based mutagenesis, had biased induction of mutations in A and T, low coverage of mutation types, excessive stop codons, or excessive same amino acid mutations with redundant codons.

Here, we report a saturation mutagenesis approach based on mammalian cell surface display system in order to effectively optimize the existing antibodies against viruses with higher and broader neutralization capacities. Based on this approach, we sought to optimize a human neutralizing antibody REGN10987, marketed as imdevimab, which was initially isolated from convalescent patients' B cells [11]. We applied our mammalian cell screening system in vitro to optimize antibody sequences that potentially improved binding affinity against several prevalent SARS-CoV-2 variants.

## Results

### A New method for the rapid generation of a SARS-CoV-2 Antibody against Diverse RBD Variants with higher affinity

We integrated chip-based DNA oligo synthesis technology and a single cell-based assay to develop a method for the rapid generation of an antibody with higher affinity against diverse SARS-CoV-2 variants. The method comprised of four steps, including the development of a single cell-based assay, the construction of an antibody library, the screening of positive cells using fluorescence-activated cell sorting (FACS) technology and the decoding of antibody sequence using the DNA sequencing technology.

First, we developed a single cell-based assay, in which an antibody was displayed on the surface of a mammalian cell and bound to a targeted protein, such as S-RBD herein. To ensure the antibody to be assayed on the surface of mammalian cells, we adopted the structure of CAR-T, in which anti-CD19 single-chain fragment variable (scFv) was replaced with a SARS-CoV-2 neutralizing antibody, such as REGN10987 that has been authorized for emergency use (EUA) [11, 37]. In brief, the N-terminal 21-aa signal peptide and C-terminal 24-aa peptide of REGN10987 scFv were retained for the purpose of its correct transportation and transmembrane to plasma membrane. The whole scFv fragment was inserted into the 5' end of the mCherry encoding region via a T2A tag, packaged into lentivirus, and then expressed in HEK 293T cells (Fig 1A). After 48h, the cells were treated with S-RBD conjugated with fluorescein isothiocyanate (RBD-FITC), and examined by confocal fluorescence microscopy. The positive cells expressing mCherry (red) were labeled green (Fig 1B).

Furthermore, to assess RBD binding specificity, a dose-binding affinity curve was observed with the increase of the RBD-FITC concentration, as quantified by FACS (Fig 1C). In contrast, when treated with 10 nM RBD-FITC, the cells were competitively treated with the different concentration of non-FITC RBD, and the fluorescence signal gradually weakened (Figs 1E and S1A). Interestingly, when a few of key amino acids located in complementarity-determining regions (CDRs) of REGN10987 scFv were mutated into alanine, the binding affinity of REGN10987 scFv to RBD-FITC was weakened or even disappeared (Figs 1D and S1B). Together, all these results demonstrate the successful establishment of a single-cell based assay that allows the rapid screening of antibody Fvs according to their binding affinity to a targeted antigen, such as a SARS-CoV-2 variant.

Second, we designed and synthesized a comprehensive library of full-length variable region of heavy chain (VH, 120 aa) and light chain (VL, 110 aa) of REGN10987 scFv protein, in which each amino acid in VH and VL domain of the protein was mutated into 19 other natural amino acids. To do so, we synthesized 4,370 mutation primers on an integrated chip (Fig 2A). Through tuning the concentration of those mutation primers to 100 ng/μL in a PCR, a library of mutated REGN10987 scFv with ~2 custom-designed point mutations was efficiently

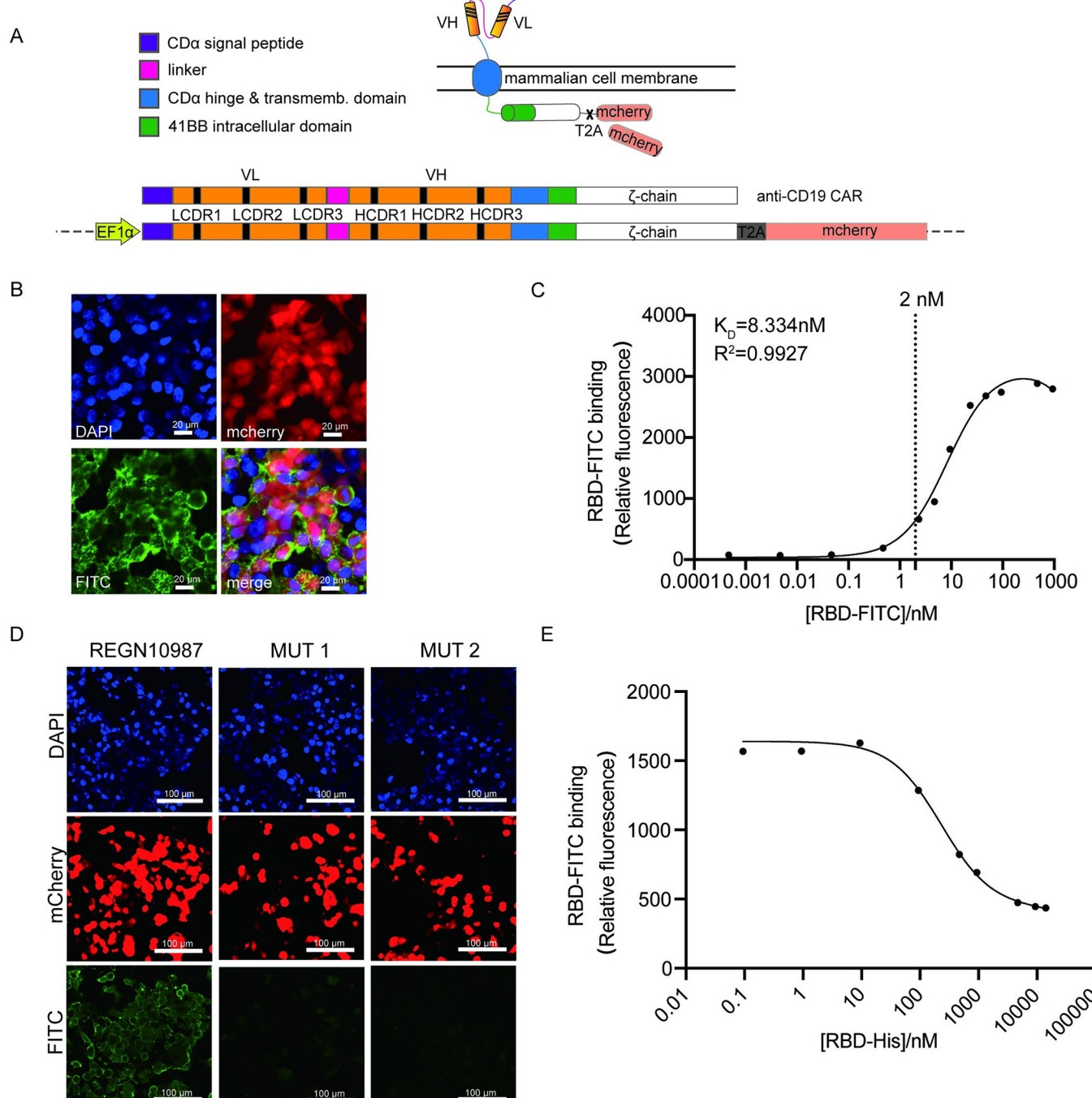

**Fig 1. Construction of an Antibody Surface Display System in HEK 293T. (A)** Schematic diagram of antibody scFv surface display system on mammalian cells. The VL and VH domain of scFv receptor construct used in anti-CD19 CAR was replaced with a SARS-CoV-2 neutralizing antibody REGN10987. VH, variable region of Ig heavy-chain; VL, variable region of Ig light-chain. **(B)** Confocal microscopic images of REGN10987 scFv displayed on HEK 293T cells. HEK 293T cells were infected by lentivirus packaged with a plasmid directing surface-expression of REGN10987 scFv. Infected cells (red) were fixed with DAPI nuclear staining followed by detection with biotinylated RBD, and then with streptavidin-FITC (green). Merged staining patterns are shown. Scale bar: 20 μm. **(C)** RBD bound to REGN10987 scFv-expressing HEK 293T cells at different titers. Concentration of RBD was from 0 nM to 1,000 nM. RBD-FITC signal was measured by flow cytometry. **(D)** Confocal microscopic images of mutated REGN10987 scFv displaying on HEK 293T cells. Several key amino acids in CDRs were mutated into alanine. Mutated amino acid was showed in S1B Fig. Transfected cells (red) were fixed with DAPI nuclear staining followed by detection with biotinylated RBD, and then with streptavidin-FITC (green). Scale bar: 100 μm. **(E)** RBD-His added to RBD-FITC competitively bound to REGN10987 scFv-expressing HEK 293T cells. Concentration of RBD-FITC was 10 nM, while different titers of RBD-His was from 0 nM to 10,000 nM. RBD-FITC signal was measured by flow cytometry.

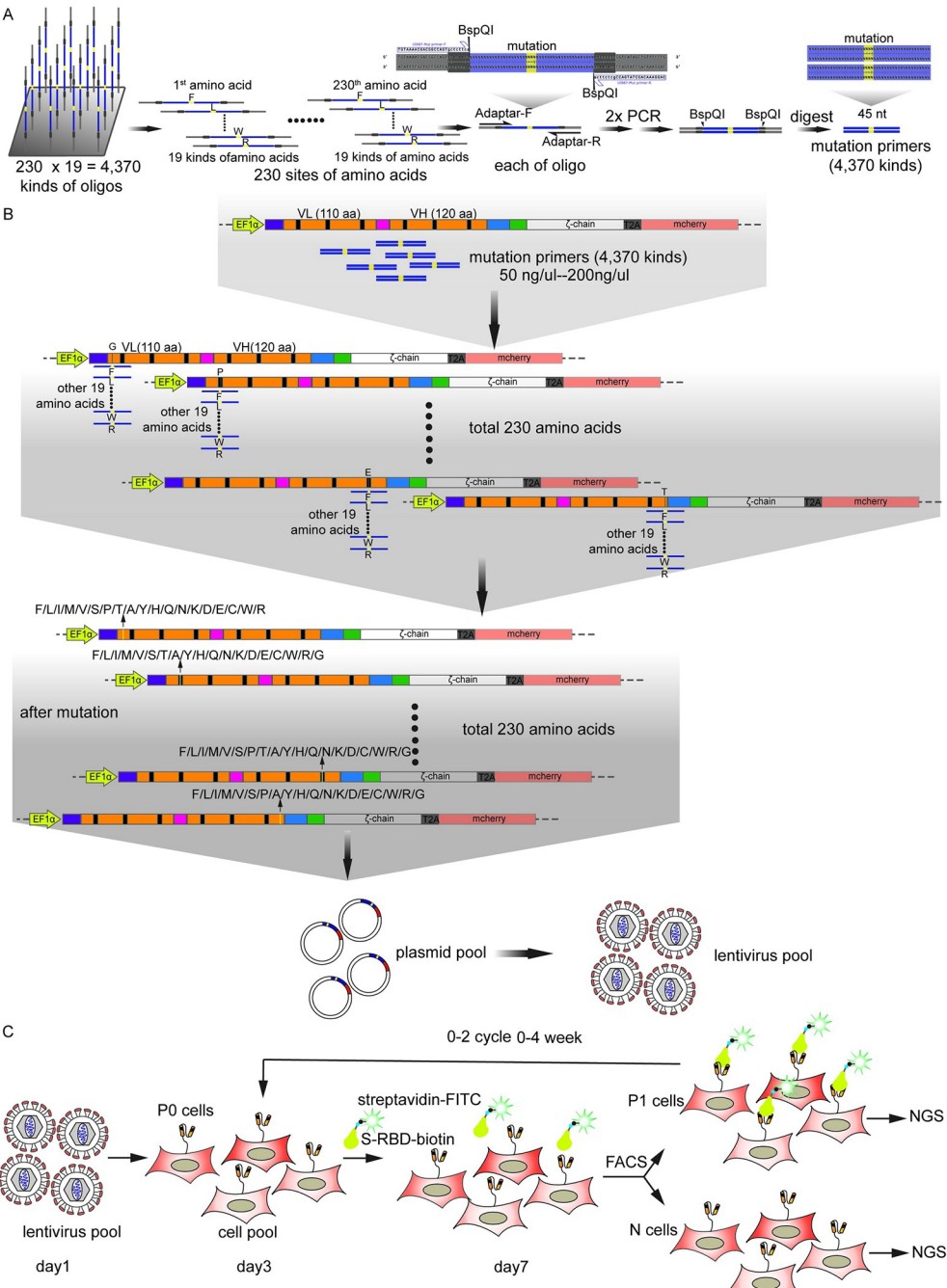

**Fig 2. High-throughput Saturation Mutagenesis Screen. (A)** A novel method for the generation of primers for high-throughput saturation mutagenesis. High-throughput chip-synthesized oligo was used to produce saturation mutagenesis primers of REGN10987 scFv. **(B)** Flow chart of obtaining REGN10987 scFv saturation mutagenesis library. High-throughput saturation mutagenesis primers were used to construct a full-length REGN10987 scFv lentiviral vector. **(C)** Timeline for establishment and screening of mutagenesis library. REGN10987 scFv vector-packaged lentivirus (MOI < 0.3) was transduced to HEK 293T cells and a mammalian cell mutant library (P0) was obtained after one round of FACS. After the incubation with SARS-CoV-2 RBD-FITC, a second round of FACS yielded RBD positive and negative groups (P1, N).

generated (Figs 2B and S2). The library was then inserted into the 5' end of the mCherry encoding region in the vector as described in Fig 1A. The constructed plasmids were transformed into *E. coli* cells and a mutant library of over $10^7$ mutants was obtained for the following screening.

Third, mutagenesis plasmid library was packaged into lentivirus and then transfected into HEK 293T cells in less than 0.3 multiplicity of infection (MOI) to guarantee no more than one copy of REGN10987 scFv variant per cell. This original cell library was defined as P0 (pre-screening) group and used for normalization later. After 48 hours, the mutant library of $10^7$ HEK 293T cells was incubated with 2 nM wild type RBD-FITC. The top 1.1% FITC-positive cells (defined as P1 group) and the top 5% FITC-negative cells (defined as N group) were collected according to the fluorescence signal density, respectively. Then, P1 group cells were cultured for several days and divided into two parts, and one half was used for the second round of selection and another half for later sequencing (Fig 2C). The top 10.1% FITC-positive cells (defined as P2 group) were collected in the second-round screening (S3A Fig). If needed, more rounds of selection could be performed.

Finally, we used the next generation of DNA sequencing technology to decode the enriched REGN10987 scFv variants. In brief, total RNAs were extracted from P0, P1, P2 and N groups of cells, respectively and used for the next generation sequencing. Biological replicates (n = 4) were conducted to avoid deviation. All of the designed mutations (4,370) were observed in P0 group. The enrichment scores of nonsynonymous mutations were set as the $\log_2$-scaled ratio of the frequencies of transcripts between the sorted populations (P1/P0, P2/P0 and N/P0, respectively).

The RBD variants initially screened also include the one of SARS-CoV-2 Beta variant (Lineage B.1.351), which have N501Y, K417N and E484K mutations. The top 1.41% FITC-positive cells (defined as BP1 group) were collected and followed by deep sequencing. The pre-screening cell library was also sequenced (defined as BP0 group) (S3 Fig). Delta S1 (Lineage B.1.617.2)-FITC probe, containing L452R and E484Q in RBD and D614G in S1, was used in the single-cell assay. Until the third round, the FITC-positive cells (defined as DP3 group) clustered distinctly (S4 Fig) due to the low binding ability of REGN10987 to Delta S1 protein. All of FITC-positive cells in three rounds and its pre-screening cell library, which were defined as DP1, DP2, DP3 and DP0 group, respectively, were deeply sequenced.

### The identification and validation of a new antibody with greater neutralization efficiency against Wild-type, Beta and Delta SARS-CoV-2 Variants

The enrichment scores of all collected 4,370 mutations were aligned along position to generate a full-length VH and VL enrichment heatmap against WT, Beta and Delta variants, respectively (Fig 3A). WT and Beta RBD bound relatively strongly to REGN10987 compared to Delta S1. A distinct clustering could also be seen in the REGN10987 scFv mutant library (S3A and S4 Figs). From full-length enrichment heatmap against either WT or Beta RBD variants, it showed that the majority of amino acid mutations in the CDRs region were deleterious and could not be enriched in the high-affinity group. This finding was in line with the complementary pairing function of the CDR regions, especially CDR3 of VH, with the antigen (Fig 3A). From enrichment heatmap of CDRs, several amino-acid mutations were enriched in the regions of LCDR3 and HCDR2, relatively (Fig 3B). Consistently, as shown in heatmaps for the screening of Delta S1, large areas of irrelevant enrichment appeared in the first and second rounds, but disappeared after the third round of screening (S4 Fig). We used relative fold change (FC) of transcript (the ratio of mean frequency after screening to mean frequency

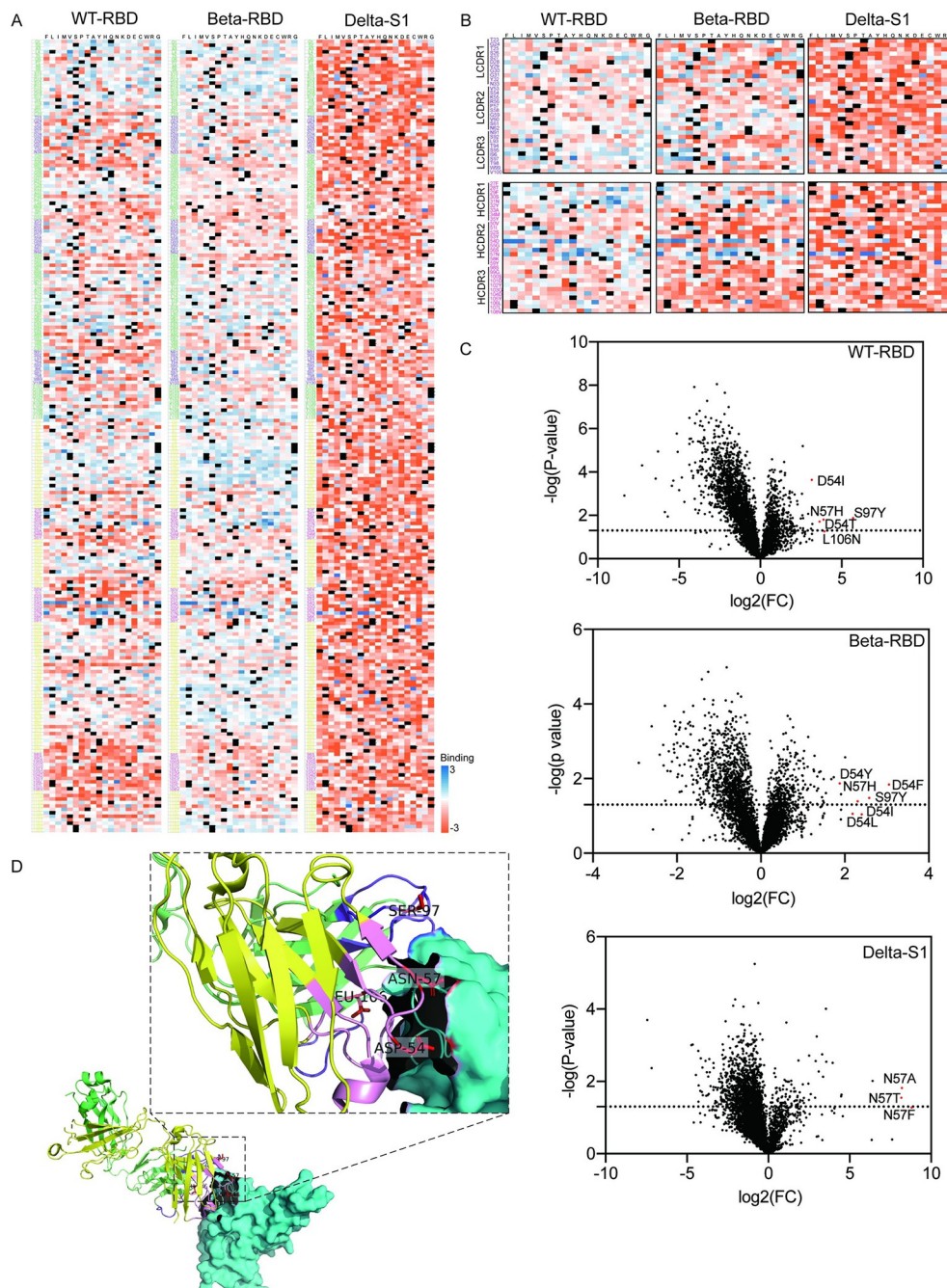

**Fig 3. Acquisition of Mutant scFv With a Higher Affinity against Wild-type, Beta and Delta SARS-CoV-2 using a HEK 293T Antibody Surface Display Library. (A)** REGN10987 scFv saturated mutagenesis heatmap of affinity binding to WT-RBD, Beta-RBD and Delta-S1. Fitness scores based on the average $Log_2$ enrichment ratios from several replications of the WT-RBD (P2/P0), Beta-RBD (BP/BP0) and Delta-S1 (DP3/DP0) sorts were plotted from depletion or deleterious (orange) to enriched (blue). Positions on REGN10987 scFv were shown on the horizontal-vertical axis, and amino acid substitutions were indicated on the vertical axis. VL and VH of REGN10987 were shown in light green and yellow, respectively. CDRs of VL were shown in purple and CDRs of VH were shown in pink. Four replications for WT-RBD, three replications for Beta-RBD, two replications for Delta-S1. **(B)** Magnified views of CDR positions of VL (upper) and VH (lower). Positions on REGN10987 scFv were shown on the horizontal-vertical axis, and amino acid substitutions were indicated on the vertical axis. CDRs of VL were shown in purple and CDRs of VH were shown in pink. **(C)** Volcano plots of increased and decreased amino acid mutations in binding affinity to WT-RBD (P2/P0), Beta-RBD (BP/BP0) and Delta-S1 (DP3/DP0). Representative mutations were highlighted in red. Data were generated from n = four independent experiments for WT-RBD, three for Beta-RBD, two for Delta-S1. FC: relative fold change

(mean frequency (after screening)/mean frequency (before screening)). The p-value was calculated using a two-sided Student's t-test. **(D)** Structure of screened mutation sites with high affinity on REGN10987. Structure (PDB: 6XDG) of REGN10987 bound to RBD (cyan). VL and VH of REGN10987 were shown in light green and yellow, respectively. CDRs of VL were shown in purple and CDRs of VH were shown in pink. S97, D54, N57 and L106 were labelled in red.

before screening) to determine the ability of single amino acid mutation affecting binding affinity.

In total, 1,541 amino acid mutations of WT RBD and 1,868 amino acid mutations of Beta RBD had FC > 1(Fig 3C). Next, we picked seven single amino acid substitution mutations that scored the highest FCs of P2/P0 or BP/BP0, including S97Y, D54T, N57H, D54F, D54I, D54L and D54Y. Among these seven mutation sites, S97 was located in CDR3 of VL, while both D54 and N57 were included in CDR2 of VH. For Delta variant S1, 929 amino acid mutations had FC > 1 in the third round (Fig 3C). Similarly, we picked three single amino acid substitution mutations scored the highest FCs of DP3/DP0, such as N572A, N57T and N57F of HCDR2. To ensure that the binding affinity was improved for both WT and Delta SARS-CoV-2, an additional substitution mutation L106N was picked from the results of WT RBD, which was located in VH's CDR3 that is believed to be an important recognition region for antigen recognition. Four amino acids (S97, D54, N57 and L106) we picked were located on the interaction binding plane of REGN10987 and RBD, so it was likely for their replacement to affect the binding affinity of antibody to RBD (Fig 3D).

Then, a group of stable cell lines each stably expressing one of REGN10987 scFv variants were generated and then incubated with RBD-FITC. Their FITC signals were quantitatively analyzed using flow cytometry. Impressively, D54F and D54Y variants showed 2–3 times higher affinity to Beta RBD than the original antibody REGN10987 (Fig 4A). Next, we generated eleven new combinatorial mutations of S97Y, D54T, N57H, D54F, D54I, D54L and D54Y, named d1-d11 (Fig 4A, right) and tested their affinity to either WT or Beta RBD, respectively. The majority of these new antibodies, such as d1, d2, d3, d6, d7, d8 and d9, showed higher affinity to both WT-RBD and Beta-RBD (Fig 4C). We selected d7 and d9, two having the highest binding affinity to both WT-RBD and Beta-RBD, for the further validation by pseudotyped virus assays.

We also generated stable cell lines and verified the binding affinity of four mutated REGN10987 scFv variants (N57F, N57T, N57A, L106N) to Delta S1 in comparison to WT RBD. All four scFv variants showed increased binding affinity to both Delta S1 and WT RBD (Fig 4B). Furthermore, we generated three combinatorial REGN10987 scFv mutations, named A1-A3. Noted, all three mutations had higher binding affinity to both Delta S1 and WT RBD, and we finally chose A3 to test neutralization using SARS-CoV-2 pseudotyped viruses (Fig 4D).

Finally, we aligned and analyzed the enrichment heatmap of FITC-negative cells (N group) for WT RBD based on FC of N/P0 (S3B Fig), and found five mutations with the highest enrichment scores, that is, C90E, A86W, C96D, Y88T and I20D (S3C Fig). Further validation showed that those five mutations hardly bound to WT RBD (S3D Fig). Together, all results from both positive and negative screening suggested that this robust method allowed us the high-throughput screening of a new antibody with higher affinity against a variety of SARS-CoV-2 variants.

## Optimized antibodies efficiently neutralize prevalent SARS-CoV-2 Pseudotyped viruses

Next, we characterized the binding affinity of d7, d9 and A3 antibodies for different RBD or S1 variants by use of surface plasmon resonance (SPR). The KD values of d7, d9 and A3

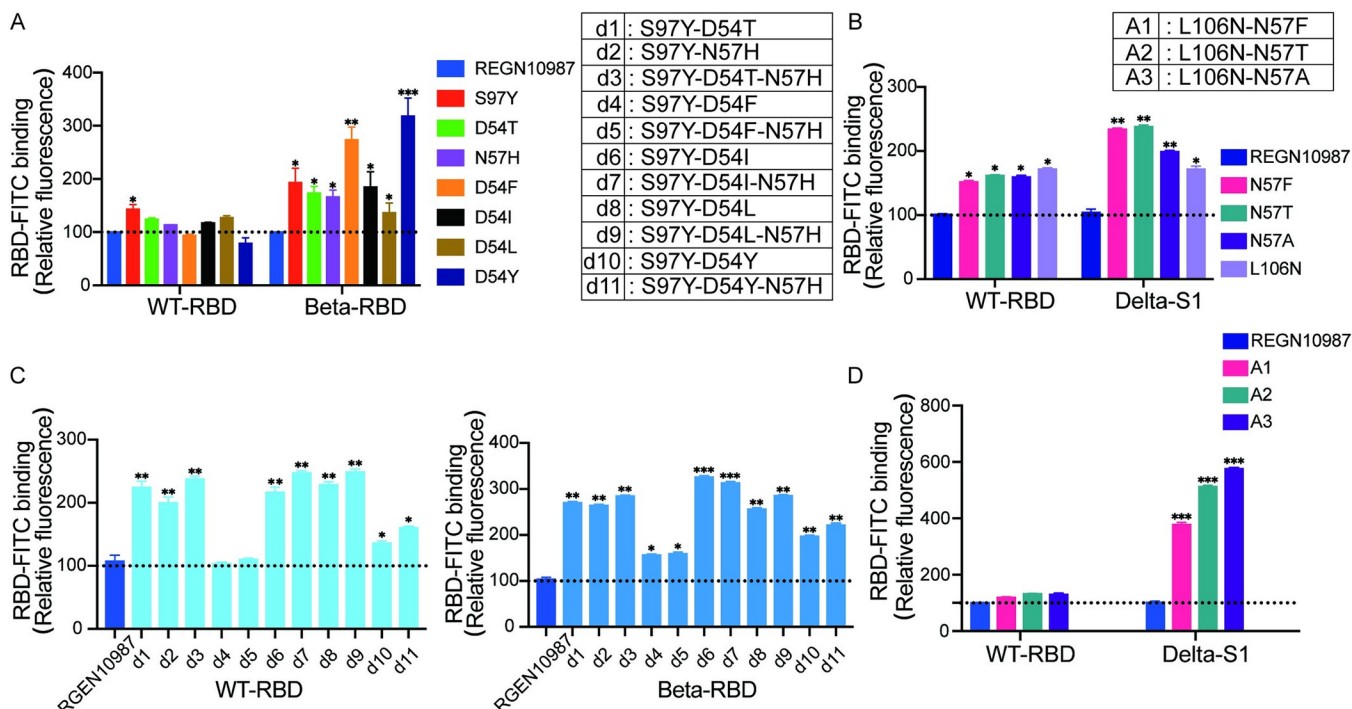

**Fig 4. Verification of Mutant scFvs using HEK 293T Antibody Surface Display System. (A)** Verification of single amino acid substitution REGN10987 scFv to WT-RBD and Beta-RBD using HEK 293T antibody surface display system (left). Data were normalized by REGN10987 scFv RBD binding fluorescence. Data are mean ± SEM, n = 2 replicates. Unpaired t test was used to analyze differences between groups. * p < 0.05, ** p < 0.01, *** p < 0.001. Right: eleven kinds of mutated combination of different single amino acid substitution. **(B)** Verification of single amino acid substitution REGN10987 scFv to WT-RBD and Delta-S1 using HEK 293T antibody surface display system (lower). Data were normalized by REGN10987 scFv RBD binding fluorescence. Data are mean ± SEM, n = 2 replicates. Unpaired t test was used to analyze differences between groups. * p < 0.05, ** p < 0.01. Upper: three kinds of mutated combination of different single amino acid substitution. **(C)** Binding of combined amino acid substitution REGN10987 scFv to WT-RBD (left) and Beta-RBD (right) using HEK 293T antibody surface display system. Data were normalized by REGN10987 scFv RBD binding fluorescence. Data are mean ± SEM, n = 2 replicates. Unpaired t test was used to analyze differences between groups. * p < 0.05, ** p < 0.01, *** p < 0.001. **(D)** Binding of combined amino acid substitution REGN10987 scFv to WT-RBD (left) and Delta-S1 (right) using HEK 293T antibody surface display system. Data were normalized by REGN10987 scFv RBD binding fluorescence. Data are mean ± SEM, n = 2 replicates. Unpaired t test was used to analyze differences between groups. *** p < 0.001.

antibodies for WT-RBD were 0.15, 0.17 and 0.17 nM, respectively, which is 3 times lower than one of original antibody REGN10987. Similarly, the KD values of d7 and d9 for Beta RBD were two times lower than one of original antibody REGN10987. For Delta S1, the KD values of original antibody REGN10987 was 0.9 nM, and all three optimized antibodies showed affinity 3-fold increased to 0.3 nM. (S5 Fig).

Second, we weighted the efficacy of d7, d9 and A3 antibodies in neutralizing SARS-CoV-2 infection by use of a pseudotyped-VSV reporter assay, in which seven psedotyped viruses encoding diverse spike proteins (WT, Alpha, Beta, Gamma, Delta, Lambda or Mu, respectively) into the envelope were infected into HEK 293T cells. Impressively, all three optimized antibodies showed significantly improved neutralizing efficiency than the original antibody REGN10987 against all 7 pseudotyped viruses with average half-maximal effective inhibitory concentration (EC50) reaching 0.005 μg/mL (Figs 5 and S6). In particular, the EC50 of d7, d9 and A3 antibodies against Lambda variants was less than 0.005 μg/mL, 50 times higher than one antibody REGN10987. Meanwhile, we examined another six authorized commercially available neutralizing monoclonal antibodies in comparison with d7, d9 and A3 antibodies against seven pseudotyped viruses (Figs 5 and S6). Strikingly, all three optimized antibodies have dramatically greater neutralization efficiency for all seven tested viruses.

| EC$_{50}$(ng/mL) | S309 | VHH7.2 | LY-COV555 | JS016 | AZD1061 | REGN10933 | REGN10987 | d7 | d9 | A3 |
|---|---|---|---|---|---|---|---|---|---|---|
| WT | 298.89 | 1309.81 | 43.11 | 84.69 | 31.76 | 25.39 | 52.90 | <5 | <5 | <4 |
| Alpha | 431.77 | 1704.53 | 76.92 | 331.55 | 30.86 | 35.97 | 26.55 | <5 | <5 | <4 |
| Beta | 199.15 | 803.55 | >3333 | >3333 | 12.87 | 317.06 | 14.79 | <5 | <5 | <5 |
| Gamma | 274.69 | 946.98 | >3333 | >3333 | 30.43 | 1405.18 | 17.87 | <5 | <5 | <3 |
| Delta | 149.88 | 392.85 | 1894.49 | 24.49 | 34.30 | 7.63 | 26.88 | 5.59 | 5.84 | <5 |
| Lambda | 490.49 | 1545.61 | >3333 | 66.76 | 109.16 | 20.89 | 223.81 | <5 | <5 | <5 |
| Mu | 285.21 | 451.28 | >3333 | >3333 | 35.07 | 347.45 | 20.96 | 5.92 | 5.65 | <5 |

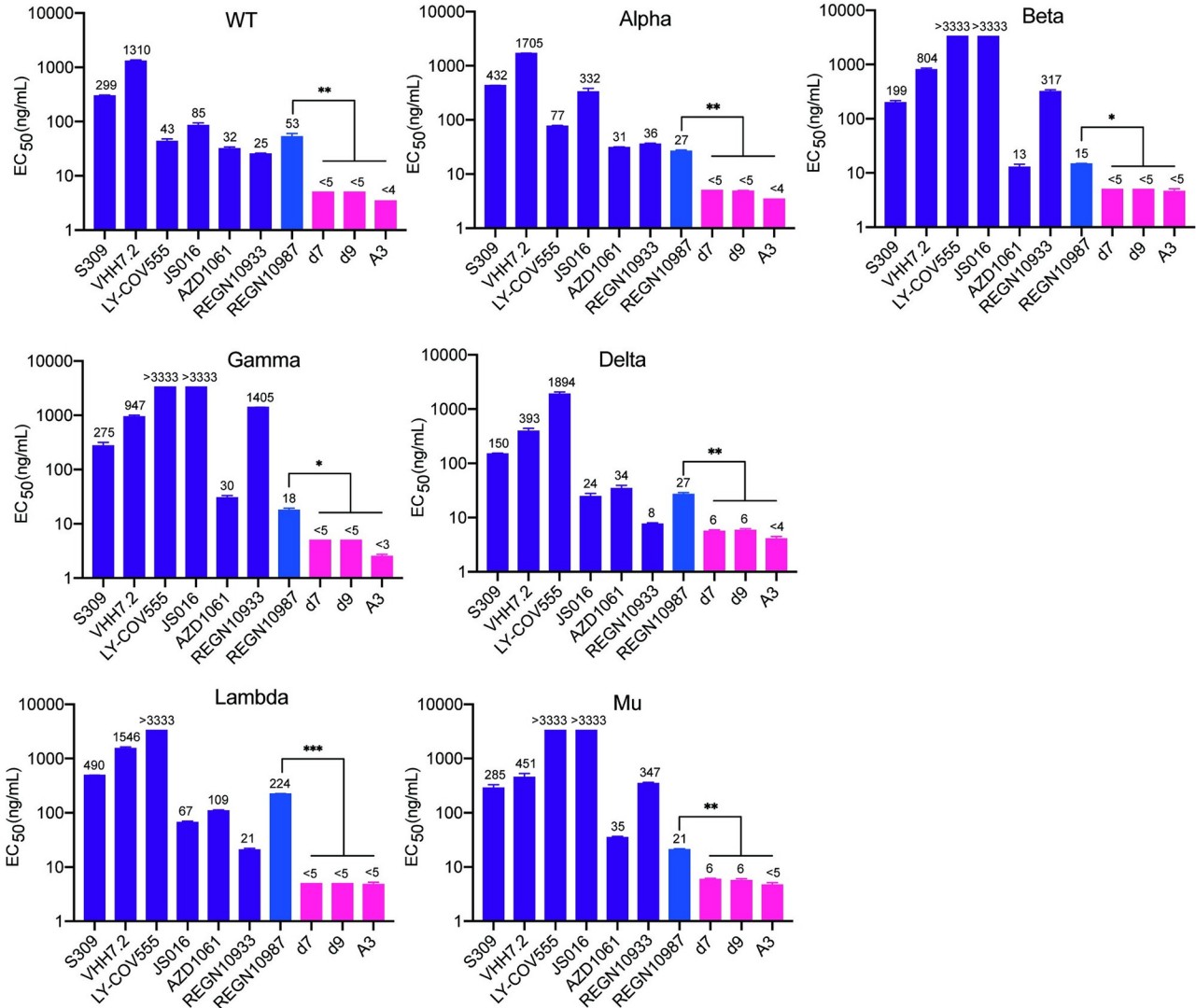

**Fig 5. Broad Efficient Neutralization level of Optimized Antibodies.** EC50 values of optimized antibodies prevalent SARS-CoV-2 spike-pseudotyped viruses. The entry of SARS-CoV-2 and its variants (Alpha, Beta, Gamma, Delta, Lambda or Mu) into cell lines expressing hACE2 was blocked with monoclonal antibodies. Unpaired t-test was used to analyze differences between groups. * $p < 0.05$, ** $p < 0.01$, *** $p < 0.001$. All results were shown as mean ± SEM. In each experiment, the infection assay was performed in duplicate wells.

## Discussion

As the virus continues to evolve, more and more variants will appear in the infected population. Many of these variants, particularly five VOCs, employ different mutating strategies for immune escape, resulting in neutralization failure of vaccines and antibodies. It is no doubt

urgent to develop a new antibody in order to overcome emerging or future variants with broader and higher neutralizing activities.

Here, we show a mammalian-cell surface display system that displays a functional antibody scFv at the cell surface and can be used as antibody affinity verification. Furthermore, this system enables the rapid construction of a combinatorial library and the high-throughput screening of antibody with higher affinity by use of flow cytometry. The selected single-chain antibodies can easily be converted to complete IgG antibodies for clinical use. As a proof-of-concept approach to antibody engineering, this display system should be an important alternative to the existing affinity-based selection techniques, such as phage display, bacterial surface display and yeast cell surface display, for either antigen-antibody binding or other biologically important protein-protein interactions.

Protein engineering by directed evolution is an important way to optimize protein and plays a significant role in life science research [38, 39]. In addition to error-prone PCR, both mutation bias and low coverage of amino acid mutations and degenerate codons (NNN, NNS, NNK) have been widely used to obtain saturated protein mutations [40–42]. However, those approaches have many inherent problems, including codon redundancy, uneven distribution of amino acid mutations and nonsense mutations. Recent advances in DNA synthesis techniques, such as oligonucleotide microarrays that allow the non-degenerate saturation mutagenesis of contiguous codons, are expected to solve these problems well [43, 44]. To overcome the above limitations, we constructed a scFv saturated mutagenesis library based on SARS-CoV-2 neutralized monoclonal antibody REGN10987 using a precisely designed oligonucleotide pool. Saturated mutations were obtained for each amino acid in 230 amino acid sequences, with coverage of 100% of all expected variations. In addition, mutagenesis can be introduced at multiple sites by adjusting the concentration of mutation primers, further increasing the diversity of the mutagenesis library. Therefore, our method achieves saturation mutagenesis with rational design, achieving more comprehensive variant coverage and efficiency.

Based on our mutation and optimization approach, we obtain several optimized REGN10987 antibodies and demonstrate their remarkable neutralization capacities against diverse prevalent SARS-CoV-2 mutants. The high-affinity mutant described here would provide a new idea in the treatment of patients infected with either of those seven SARS-CoV-2 variants. At the same time, we got optimized antibodies d7 (S97Y, D54I, N57H) and d9 (S97Y, D54L, N57H) from the screening results of WT-RBD and Beta-RBD, while A3 (N57A, L106N) from WT-RBD and Delta-S1. It proves the importance of the four sites S97, D54, N57 and L106 on the REGN10987 antibody to bind the SARS-CoV-2 virus. Although screened by different RBD or S1 variants strains, the neutralizing activity of these three optimized antibodies against seven popular mutant strains is greatly improved. This is probably because the change in the N57 site, included in all optimized antibodies, may play a key role in antigen recognition and binding.

The SARS-CoV-2 virus is still constantly mutating. Generally, it takes much time and effort to separate and validate effective antibodies from the serum of recovered patients infected with a newly emerged mutant virus. Indeed, it is also difficult to obtain and handle patients' serum without extra professional equipment and skill. Nevertheless, our mammalian cell surface display system can be operated in ordinary biological laboratories. At this time, our antibody optimization approach can take advantage of its simple and fast features, and optimize the existing antibodies with better effects. The same is true for mutated viruses that may appear in other species, such as monkey pox. In addition, this simple and efficient mammalian cell screening system can also be easily implemented to engineer other antibodies or proteins in the field of cancer immunotherapies. Meanwhile, it would become more accurate and faster to

screen high-affinity antibodies if our approach could be combined with feedback control and other AI based strategies.

## Materials and methods

### Plasmids

The anti-CD19 scFv was replaced with antibody REGN10987 (PDB ID 6XDG) scFv region and inserted into the 5' end of the mCherry encoding region via a T2A tag. The whole fragment was inserted into NotI and EcoRI sites of a lentivirus transfer vector pLV03 by Gibson Assembly to obtain the plasmid pLV03 REGN10987.

### Cell culture and cell lines

HEK 293T cells were cultured in Dulbecco Modified Eagle Medium (DMEM, Thermo Fisher) supplemented with 10% fetal bovine serum (FBS, Hyclone-Thermo Fisher) and 1 × penicillin-streptomycin (Gibco) at 37˚C, 5% $CO_2$. Cell culture media and supplements were obtained from Life Technologies, Inc. Stable cell lines with single or multiple mutated sites were obtained by lentivirus infection with the plasmids containing the corresponding mutated sites (multiplicity of infection [MOI] less than 0.3).

### Proteins and monoclonal antibodies

SARS-CoV-2 WT spike RBD, SARS-CoV-2 Beta variant RBD, SARS-CoV-2 Delta variant S1 proteins were purchased from Sino Biological, Inc. Purified monoclonal antibody d7, d9 and A3 were ordered according to amino acid sequences from Suzhou SGE Biotech. A total of seven neutralizing monoclonal antibodies against SARS-CoV-2 S protein were purchased from Wuhan Chemstan Biotechnology Co., Ltd.

### Confocal fluorescence microscopy

HEK 293T cells infected with pLV03 REGN10987 were grown on cover slips. After 48 hours, cells were washed with PBS and fixed in 1 mL of 4% formaldehyde in PBS for 10 minutes at 25˚C. Cells were blocked with 5% BSA in PBS (Hyclone) for 1 hour at 25˚C and then incubated in 1% FBS DMEM with 0.05 μg/mL of biotinylated recombinant spike RBD. After incubation for 30 min at 37˚C, cover slips were washed PBS-BSA, incubated with 1000 × streptavidin-FITC (Biolegend) diluted in 1% FBS DMEM for another 30 minutes at 37˚C, and then washed. To visualize the nucleus, cells were stained with DAPI (Molecular Probes) for 5 min at 25˚C. Slides were analyzed in Nikon A1R Super high resolution confocal microscope.

### REGN10987 scFv saturation mutagenesis

230 positions (120 aa of VH and 110 aa of VL) on REGN10987 scFv were mutated to the other 19 amino acids based on the REGN10987 scFv coding sequence. The mutated codon was the most commonly used human codon per amino acid. Every mutagenesis primer was 45 nt contained the mutated codon and the 21 nucleotides upstream and downstream. Recognition site for BspQI restriction endonuclease was at both end of the 21 nucleotides followed by a 18 nt adaptor for amplification. A high-throughput oligonucleotide synthesizer (CustomArray B3) was used to synthesize the oligonucleotides on a 12k microreactor chip. Then the oligonucleotides were collected according to manufacturer's instructions.

The oligo pool was amplified by PCR using KAPA HiFi HotStart ReadyMix DNA Polymerase (Roche). PCR products were purified after electrophoresis and then cut with BspQI restriction endonuclease. The 45-nt mutagenesis primers were finally obtained through gel

electrophoresis and purification. The collected mutagenesis primers were added to the pLV03 REGN10987 plasmid, amplified with KAPA HiFi HotStart DNA Polymerase, and assembled to pLV03 REGN10987 by Gibson assembly. To establish the REGN10987 antibody scFv mutagenesis library, the generated circular plasmid was packaged into a lentivirus and infected to HEK 293T cell with MOI less than 0.3. 48 hours after infection, $1 \times 10^7$ mCherry positive cells were obtained as a cell library through FACS.

### Flow cytometry

HEK 293T Cells were analyzed 48h after lentivirus infection by flow cytometry. To bind RBD to antibody REGN10987 scFv, REGN10987 scFv-expressing cells were washed with PBS-BSA and incubated with medium containing biotinylated recombinant spike RBD or S1 protein (SARS-CoV-2 RBD, SARS-CoV-2 Beta RBD, SARS-CoV-2 Delta S1, Sino Biological Inc.) for 30 minutes at 37˚C. Cells were washed with PBS-BSA and then incubated with $1000 \times$ streptavidin Fluorescein Isothiocyanate (FITC) (Biolegend) for another 30 minutes at 37˚C. Finally, cells were washed with PBS-BSA and analyzed on a Beckman Astrios EQ. Data was processed with FlowJo.

### Data curation and analysis

Total RNA was extracted from P0, P1, P2, N, BP0, BP1, DP0, DP1, DP2, DP3 groups with TRNzol reagent (TIANGEN), and cDNA was reverse transcribed using PrimerScript RT reagent Kit (TaKaRa). The coding sequences of antibody REGN10987 scFv were amplified as 4 fragments. Adapters were added to the ends of the products by flanking sequences on the primers for annealing to Illumina sequencing primers. Amplicons were sequenced by using a $2 \times 250$ nt paired end protocol on an Illumina NovaSeq 6000. Two to four groups of biological parallel experiments were conducted to avoid deviation. The raw variant read count data was collected and the enrichment scores were calculated by $\log_2$ of FC. FC = mean frequency (after screening)/mean frequency (before screening), such as P1/P0, P2/P0, N/P0.

### Antibody REGN10987 scFv Mutant Cell Lines binding to Wild Type or Variant spike proteins

Mutant antibody REGN10987 scFvs were packaged as lentivirus and infected to HEK 293T cells. Positive cells were collected through Flow Cytometry. For each antibody mutant, cells were digested, incubated for 30 minutes in DMEM with Wild type or variant biotinylated spike proteins. Supernatants were removed by centrifuge. Precipitates were washed once with PBS-BSA, then added $1000 \times$ streptavidin-FITC (Biolegend) diluted in 1% FBS DMEM. After 30 minutes of incubation, supernatants were removed by centrifuge. Precipitated cells were washed with PBS-BSA and analyzed using LSRFortessa Cell Analyzer (BD).

### Surface plasmon resonance test for antibody binding

The affinity between monoclonal antibody and SARS-CoV-2 Wild Type or Variant Spike RBD (Sino Biological Inc.) was quantitively measured using Biacore 2000 instrument. RBD (20 ng/μL) was dissolved in sodium acetate buffer (PH = 4.5) and immobilized as the ligand on a CM5 chip at a level of 60 response units (RUs). $1 \times$ HBS-EP (Cytiva) was used as running buffer. Several kinds of mutated and original antibody REGN10987 were employed as the analyte, with concentrations serially diluted with running buffer from 100 to 1.56 nM. Flow rate was set at 45 μL/min, with 180 s for binding and 1800 s for dissociation. The sensor chip was

regenerated with imidazole for 30 s at the end of a running circle. The resulting data were fit to 1:1 binding model with the Biacore evaluation software

## Pseudotyped neutralization assay

Pseudotyped viruses of SARS-CoV-2 and its variants were constructed according to previous studies [45, 46]. The HEK 293T cells were adjusted to concentrations of $5–7 \times 10^5$ cells/mL the day before transfection. Cells in 15 mL of medium were transferred to a T75 cell culture flask and incubated overnight at 37˚C with 5% $CO_2$. When the cells reached 70%–90% of confluence, the culture medium was discarded and 15 mL of G*ΔG-VSV virus (VSV-G pseudotyped virus, Kerafast) at a concentration of $7 \times 10^4$ TCID50/mL was used for infection. At the same time, the cells were transfected with 30 µg of S protein expression plasmid according to the instructions of Lipofectamine 3000 (Invitrogen) and then cultured with 5% $CO_2$ at 37˚C in an incubator. The cell supernatant was discarded after 6–8 hours and gently washed twice with 1% FBS-PBS. Then, the cells were added with 15 mL fresh complete DMEM and incubated at 37˚C with 5% $CO_2$ for 24 hours. Cell supernatant containing pseudotyped virus was harvested and centrifuged at $1000 \times g$ for 10 min, filtered with 0.45-µm filter, aliquoted, and frozen at -80˚C for further use.

The evaluation of infection inhibition effect of the original antibody REGN10987 and mutated monoclonal antibodies was detected by the reduction of luciferase gene expression in the pseudotyped virus infection assay. The test samples (antibodies) were serially diluted using complete DMEM culture media in a 96-well plate for a total of seven gradients, (starting with 300 nM and three-time gradient), after which 50 µL of virus solution was added. On each 96-well plate, seven cell control wells (only a complete culture medium was added) and seven virus control wells (no test sample but only virus solution was added) were arranged. The 96-well plates were incubated at 37˚C and 5% $CO_2$ for 1 hour, after which trypsin-treated (0.25% Trypsin-EDTA, Gibco) ACE2 expressing HEK 293T cells ($2 \times 10^4$/100 µL) were added to each well. After incubation at 37˚C and 5% $CO_2$ for 24 hours, 100 µL of luciferase substrate (PerkinElmer) was added to each well. 150 µL of lysate was mixed and transferred to a white opaque 96-well microplate (PerkinElmer) and measured using a luminometer (PerkinElmer), after incubation at room temperature for 2 min. Each group of antibody samples contained two replicates. The EC50 value of each sample was calculated using the Reed–Muench method.

## Supporting information

**S1 Fig. Verification of the Antibody Surface Display System in HEK 293T.** (A) Confocal microscopic images of RBD-His added to RBD-FITC competitively bound to REGN10987 scFv-expressing HEK 293T cells. Upper: HEK 293T cells expressed REGN10987 scFv. Lower: WT HEK 293T cells. Concentration of RBD-FITC was 10nM, while different titers of RBD-His was from 0 nM to 100 nM. Transfected cells were fixed with DAPI nuclear staining followed by detection with biotinylated RBD, followed by Streptavidin-FITC (green). Scale bar: 100 µm. (B) Four amino acids of key sites mutated into alanine in CDRs. Left: Sanger sequencing showed amino acid mutations, one sites in HCDR1 and three sites in HCDR3. Right: FITC signal for REGN10987 scFv and mutated scFvs displayed on HEK 293T cells binding with RBD-FITC after mutated key sites into alanine in CDRs. MUT1 contained three mutated amino acids in HCDR3, while MUT2 contained all of four mutated amino acids. Unpaired t test was used to analyze differences between groups. $^*$ p < 0.05, $^{**}$ p < 0.01. (TIF)

**S2 Fig. Quantity Control of Mutated Amino Acids in REGN10987 scFv Saturation Mutagenesis Library.** Ratios of amino acid mutations on REGN10987 scFv per plasmid based on a Sanger sequencing result of 32 pLV03 REGN10987 scFv mutation library plasmids. Concentration gradient of mutation primers from 50 ng/μL to 200 ng/μL was added in a PCR system. (TIF)

**S3 Fig. Screening of either Higher Binding Affinity or Non-affinity scFv against Wild-type or Beta SARS-CoV-2.** (A) FACS imaging for the cell mutant libraries in different screening rounds of WT-RBD (upper) and Beta-RBD (lower). P0 referred to the original cell library before screening. P1 and N referred to FITC-positive and FITC-negative cells respectively in the first round of screening by WT-RBD, and P2 referred to FITC-positive cells in the second round. BP referred to FITC-positive cells in the first round of screening by Beta-RBD. (B) REGN10987 scFv saturated mutagenesis heatmap of non-affinity binding to WT-RBD. Fitness scores based on the average $Log_2$ enrichment ratios from two replications of the WT-RBD (N/P0) sorts were plotted from depletion or deleterious (orange) to enriched (blue). Positions on REGN10987 scFv were shown on the horizontal-vertical axis, and amino acid substitutions were indicated on the vertical axis. VL and VH of REGN10987 were shown in light green and yellow, respectively. CDRs of VL were shown in purple and CDRs of VH were shown in pink. (C) Volcano plots of increased and decreased amino acid mutations in binding affinity to WT-RBD of N groups. Representative mutations were highlighted in red. FC: relative fold change (mean frequency (after screening)/mean frequency (before screening)). Data were generated from n = 2 independent experiments. The p-value was calculated using a two-sided Student's t-test. (D) Verification of single amino acid substitution REGN10987 scFv to WT-RBD using HEK 293T antibody surface display system. Data were normalized by REGN10987 scFv RBD binding fluorescence. Data are mean ± SEM, n = 2 replicates. Unpaired t test was used to analyze differences between groups. $^*$ p < 0.05, $^{**}$ p < 0.01. (TIF)

**S4 Fig. Screening of Higher Binding Affinity scFv against Delta SARS-CoV-2 for Three rounds.** Upper: FACS imaging for the cell mutant libraries in three screening rounds of Delta-S1. DP1, DP2 and DP3 referred to FITC-positive cells respectively in the three rounds of screening. Lower: REGN10987 scFv saturated mutagenesis heatmap of high affinity binding to Delta-S1 in three screening rounds. Fitness scores based on the average $Log_2$ enrichment ratios from two replications of Delta-S1 (DP1/DP0, DP2/DP0, DP3/DP0, respectively) sorts were plotted from depletion or deleterious (orange) to enriched (blue). Positions on REGN10987 scFv protein were shown on the horizontal-vertical axis, and amino acid substitutions were indicated on the vertical axis. VL and VH of REGN10987 were shown in light green and yellow, respectively. CDRs of VL were shown in purple and CDRs of VH were shown in pink. (TIF)

**S5 Fig. Binding Affinity of Optimized Antibodies to SARS-CoV-2 Variants Spike.** Immobilized WT-RBD, Beta-RBD or Delta-S1 association (t = 0 to 180 s) and dissociation (t > 180 s) with REGN10987 (A) and optimized antibodies d7 (B), d9 (C), A3 (D) measured by surface plasmon resonance (SPR). (TIF)

**S6 Fig. Neutralization Abilities Measured by seven prevalent SARS-CoV-2 Spike-pseudotyped VSV Neutralization Assays, Related to Fig 4.** (TIF)

## Acknowledgments

We thank Institute of Molecular Medicine and National Centre for Protein Science at Peking University for technical help on flow cytometry, confocal microscopy and surface plasmon resonance. We acknowledge the funding support from "Laboratory for Synthetic Chemistry and Chemical Biology" under the Health@InnoHK Program launched by Innovation and Technology Commission, The Government of Hong Kong Special Administrative Region of the People's Republic of China.

## Author Contributions

**Conceptualization:** Jianzhong Jeff Xi.

**Data curation:** Ye Yang, Shuo Liu, Yufeng Luo, Bolun Wang, Junyi Wang, Juan Li, Jiaxin Li.

**Formal analysis:** Ye Yang.

**Funding acquisition:** Youchun Wang, Jianzhong Jeff Xi.

**Investigation:** Ye Yang, Shuo Liu.

**Methodology:** Ye Yang.

**Project administration:** Buqing Ye, Jianzhong Jeff Xi.

**Resources:** Youchun Wang, Jianzhong Jeff Xi.

**Software:** Junyi Wang.

**Supervision:** Youchun Wang, Jianzhong Jeff Xi.

**Validation:** Ye Yang, Shuo Liu.

**Visualization:** Ye Yang, Jianzhong Jeff Xi.

**Writing – original draft:** Ye Yang, Jianzhong Jeff Xi.

**Writing – review & editing:** Ye Yang, Jianzhong Jeff Xi.

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
