## [Decision Letter · Decision Letter 0]

8 Nov 2022

Dear Professor Xi,

Thank you very much for submitting your manuscript "High-throughput saturation mutagenesis generates a high-affinity antibody against SARS-CoV-2 variants using protein surface isplay assay on a human cell" for consideration at PLOS Pathogens. As with all papers reviewed by the journal, your manuscript was reviewed by members of the editorial board and by several independent reviewers. The reviewers appreciated the attention to an important topic. Based on the reviews, we are likely to accept this manuscript for publication, providing that you modify the manuscript according to the review recommendations.

Sincerely,

Katie J Doores

Academic Editor

PLOS Pathogens

Kanta Subbarao

Section Editor

PLOS Pathogens

Kasturi Haldar

Editor-in-Chief

PLOS Pathogens

orcid.org/0000-0001-5065-158X

Michael Malim

Editor-in-Chief

PLOS Pathogens

orcid.org/0000-0002-7699-2064

Reviewer Comments (if any, and for reference):

Reviewer's Responses to Questions

**Part I - Summary**

Reviewer #1: Review report for “High-throughput saturation mutagenesis generates a high-affinity antibody against SARS-CoV-2 variants using protein surface isplay assay on a human cell” by Yang et al submitted to PLoS Pathogens.

This study by JZ’s group and collaborators reports a high-throughput approach for generating high-affinity antibody against SARS-CoV2 variants. The strategy incorporates saturation mutagenesis, microarray-based oligonucleotide synthesis, and single-cell screening assay to generate a group of large set of antibodies against diverse prevalent SARS-COV-2 variants based on the human antibody REGN10987. The process can be done within 2 weeks and improve the affinity of antibodies with EC50 below 5 ng/ml.

Overall, this is a very interesting and timely article. As most countries have removed the pandemic regulations, the incidence rate of COVID-19 has been sky-high. Even though the hospitalization and severe disease rates are not as high in the current variants that are circulating. The actual amount remain alarming, especially for regions that have a low vaccination. The development of potent neutralizing antibodies is critically important in the coming time. Technologically, the approach sounds and presents a systems strategy for targeting the ever evolving COVID variants. The short turnaround and significant enhancement in affinity for broad variants are very impressive. The writing is overall good, and the data are of high quality. I recommend rapid acceptance of the manuscript and notify WHO for this new development for addressing the pandemic.

Reviewer #2: The authors have described in detail a novel antibody optimization approach to obtain high-affinity antibodies against SARS-CoV-2. After reviewing the manuscript, I found that brief description of the structure of the virus and the properties of the spike (S) glycoprotein would be useful to aid in understanding of the topic, including on why the receptor binding domain (RBD) is often targeted. I would suggest including more details in the introduction on the other techniques used for antibody generation. For example, what are the authors referring when mentioning “the serious problems related to antibody activity” (pg 2, paragraph 2)?

Please ensure that “SARS-CoV-2” is spelt with the appropriate case throughout the manuscript, including the abstract, i.e. SARS-CoV-2 and not SARS-COV-2.

**Part II – Major Issues: Key Experiments Required for Acceptance**

Reviewer #1: none

Reviewer #2: (No Response)

**Part III – Minor Issues: Editorial and Data Presentation Modifications**

Reviewer #1: Here are several optional suggestions to be considered by the authors.

1) The platform appears to be high versatile and may be applicable to address other emerging concerns, such as monkey pox and other infectious diseases. It will be useful if the author to expand the discussion to discuss the potential translation of the platform other diseases.

2) The use of feedback control and other AI based strategies has recently been adopted in the personalized medicine space. Are there efforts in using AI based approaches for optimizing potent antibodies? The author may consider discussing the compatibility between the platforms.

Reviewer #2: Suggested amendments/additions below are in parantheses:

Introduction

Page 2 Paragraph 1:

“So far, hundreds of millions of people have been confirmed infections and millions confirmed deaths (World Health Organization).” The wording here is confusing, the sentence should read “So far, there have been hundreds of millions confirmed infections and millions of confirmed deaths (World Health Organization).” Is there a specific WHO reference for this as well?

“As the epidemic progress, mutated strains with immune escape ability are emerging. The most serious strain is (the) Omicron strain that carries over 30 mutations on the S protein, half of which are located in the RBD. There is striking evidence that SARS-(CoV)-2 variants, particularly Omicron, have been less sensitive or completely escaped (from) antibodies [11-20].”

Ten references were cited here – suggested to elaborate on the antibody evasion of the SARS-CoV-2 variants mentioned. Which of the variants (Omicron, and others such as Delta) had especially reduced sensitivity to neutralising antibodies, and why?

Page 3 paragraph 1:

“…which was initially isolated from convalescent patients(’) B cells.”

Methods

Under the Flow Cytometry section: “To (bind) RBD to antibody REGN10987 scFv…”

Results

Page 6 paragraph 2:

Figure 2B was mentioned here but there appears to be a typo: “Fig. 2bB” should be “Fig. 2B”.

Page 6 paragraph 4:

“Finally, we (used) the next generation of DNA sequencing technology to decode…”

“All of (the) designed mutations (4,370) were observed in P0 group.”

“If needed, more (rounds) of selection could be performed.”

Page 9 paragraph 2:

“(In total), 1,541 amino acid mutations of WT RBD and 1,868 amino acid mutations…”

“To ensure that the binding affinity was (improved for) both WT and Delta SARS-CoV-2, an additional substitution mutation L106N was picked from the results of WT RBD, which was located in VH’s CDR3 that is believed (to be) an important recognition region for antigen recognition.”

Discussion

Page 15 paragraph 1:

“As the virus continues to evolve, more and more variants (will) appear in the infected population.”

What are examples of the variants of concern (VOC) mentioned here, and what are the mutating strategies that they employ for immune escape? Suggest to elaborate further.

Page 15 paragraph 3:

Authors mentioned “Recent advances in DNA synthesis techniques are expected to solve these problems well [40, 41]”. What advances in DNA synthesis techniques are the authors referring to?

With the use of various techniques, what is the feasibility of using the described mammalian-cell surface display system in lower resource settings? Perhaps this can be discussed as well.

Page 16 paragraph 1:

“This is probably because the change (in the) N57 site, included in all optimized antibodies, may play a key role in antigen recognition and binding.”

PLOS authors have the option to publish the peer review history of their article (what does this mean?). If published, this will include your full peer review and any attached files.

Reviewer #1: No

Reviewer #2: No

Figure Files:

Data Requirements:

Reproducibility:

References:

---

## [Editor Report · Decision Letter 1]

12 Jan 2023

Dear Professor Xi,

We are pleased to inform you that your manuscript 'High-throughput saturation mutagenesis generates a high-affinity antibody against SARS-CoV-2 variants using protein surface display assay on a human cell' has been provisionally accepted for publication in PLOS Pathogens.

Best regards,

Katie J Doores

Academic Editor

PLOS Pathogens

Kanta Subbarao

Section Editor

PLOS Pathogens

Kasturi Haldar

Editor-in-Chief

PLOS Pathogens

orcid.org/0000-0001-5065-158X

Michael Malim

Editor-in-Chief

PLOS Pathogens

orcid.org/0000-0002-7699-2064
---

## [Editor Report · Acceptance letter]

18 Jan 2023

Dear Professor Xi,

We are delighted to inform you that your manuscript, "High-throughput saturation mutagenesis generates a high-affinity antibody against SARS-CoV-2 variants using protein surface display assay on a human cell," has been formally accepted for publication in PLOS Pathogens.

Best regards,

Kasturi Haldar

Editor-in-Chief

PLOS Pathogens

orcid.org/0000-0001-5065-158X

Michael Malim

Editor-in-Chief

PLOS Pathogens

orcid.org/0000-0002-7699-2064